(GIGA)bYte

DATA RELEASE

# A dataset of small-mammal detections in West Africa and their associated micro-organisms

David Simons[1,2,3,*], Lauren A. Attfield[2,4], Kate E. Jones[2], Deborah Watson-Jones[3,5] and Richard Kock[1]

1 The Royal Veterinary College, London, UK
2 University College London, London, UK
3 The London School of Hygiene and Tropical Medicine, London, UK
4 Imperial College London, London, UK
5 Mwanza Intervention Trials Unit, National Institute for Medical Research, Mwanza, Tanzania

## ABSTRACT

Rodents, a globally distributed and ecologically important mammalian order, serve as hosts for various zoonotic pathogens. However, sampling of rodents and their pathogens suffers from taxonomic and spatial biases. This affects consolidated databases, such as IUCN and GBIF, limiting inference regarding the spillover hazard of zoonotic pathogens into human populations. Here, we synthesised data from 127 rodent trapping studies conducted in 14 West African countries between 1964 and 2022. We combined occurrence data with pathogen screening results to produce a dataset containing detection/non-detection data for 65,628 individual small mammals identified to the species level from at least 1,611 trapping sites. We also included 32 microorganisms, identified to the species or genus levels, that are known or potential pathogens. The dataset is formatted to Darwin Core Standard with associated metadata. This dataset can mitigate spatial and taxonomic biases of current databases, improving understanding of rodent-associated zoonotic pathogen spillover across West Africa.

**Subjects** Ecology, Biodiversity, Taxonomy

**Submitted:** 07 March 2023

* Corresponding author. E-mail: dsimons19@rvc.ac.uk

Preprint submitted at https://doi.org/10.32388/ZB04GW

Included in the series: *Vectors of human disease* (https://doi.org/10.46471/GIGABYTE_SERIES_0002)

## DATA DESCRIPTION

### Context

Rodents are a diverse, globally distributed and ecologically important order of mammals. Along with bats, these two orders are believed to harbour the greatest number of host species from known and predicted zoonotic pathogens. Of the 2,220 extant rodent species, 244 (10.7%) are described as hosts of 85 zoonoses [1]. Importantly, rodent hosts of zoonoses are typically synanthropic and thrive in anthropogenically disturbed habitats, leading to a spatially heterogeneous risk of pathogen transmission [2]. Rodent-associated endemic zoonoses are a significant public health threat across much of West Africa and include bacterial, viral and protozoan pathogens. Important endemic rodent-borne zoonoses in West Africa include Lassa fever (caused by *Lassa mammarenavirus*), Leptospirosis (caused by *Leptosira* sp.) and Toxoplasmosis (caused by *Toxoplasma gondii*) [3–5]. Likely, other zoonoses are circulating in the rodent populations of this region that have not yet been described or identified as causing human infections [6]. Currently, described zoonotic pathogens are generally associated with multiple rodent species. However, a single species

may be the primary reservoir; for example, *Mastomys natalensis* is considered the primary reservoir of Lassa fever, although *Lassa mammarenavirus* infection has been associated with eleven other rodent species [7]. For this reason, understanding the structure of rodent communities, their spatial distribution and their associated pathogens are vital to understanding the hazard of endemic zoonotic disease spillover and the emergence of novel zoonotic pathogens [8].

To model host occurrence, studies assessing the risk of outbreaks of endemic zoonoses and novel pathogen emergence often use consolidated datasets, such as the Global Biodiversity Information Facility (GBIF) and International Union for Conservation of Nature (IUCN) Redlist [9–11]. Despite the importance of understanding the true distribution of rodent hosts and their pathogens, curated biodiversity databases such as GBIF and IUCN are affected by taxonomic and geographical sampling biases [12, 13]. These biases can limit the interpretability of species distribution models generated to quantify the hazard of zoonotic disease spillover into human populations and guide public health interventions [14]. Rodent trapping studies are also taxonomically and spatially biased [2, 7]. Despite these biases, we found that combining data from rodent trapping studies conducted in West Africa with data from GBIF and IUCN has the potential to increase the sampled area for commonly occurring species by up to 160% and mitigate some of the effects of these biases when modelling the distribution of rodent vectors of zoonoses [7]. We found that rodent trapping studies are more likely to be conducted in locations of relatively high human population density and include data on small-mammal species that are synanthropic [7].

Our dataset, a synthesis of 127 rodent trapping studies conducted within 17 African countries (focusing on studies conducted in West Africa), can aid the development of models based on rodent reservoir occurrence. Hence, our dataset enhances the ability to estimate the potential for pathogen spillover into human populations by providing additional geographic locations of presence and absence. For example, a recent article developed a model of the risk of Lassa fever spillover based on both *M. natalensis* occurrence and pathogen prevalence in West Africa relying on 167 locations of *M. natalensis* detections [11]. Integrating our current dataset, an additional 337 locations of *M. natalensis* detections and 320 locations of non-detection could be incorporated, increasing the coverage of observations over the endemic region.

## METHODS

### Search strategy

Our dataset contains information on small mammal detections and non-detections obtained from rodent trapping studies conducted in West Africa between 1964 and 2022. Data have been extracted from published articles, biodiversity surveys and impact assessments. Studies were identified through a search conducted in Ovid MEDLINE, Web of Science (Core collection and Zoological Record), JSTOR, BioOne, African Journals Online, Global Health, and the preprint servers BioRxiv and EcoEvoRxiv using the following terms as exploded keywords:

1. Rodent OR Rodent trap*

AND

2. West Africa

We used the United Nations (UN) definition for West Africa, which includes the following countries (ISO 3166-1 alpha-2 codes are given in parentheses): Benin (BJ), Burkina Faso (BF), Cape Verde (CV), Ivory Coast (CI), Gambia (GM), Ghana (GH), Guinea (GN), Guinea-Bissau (GW), Liberia (LR), Mali (ML), Mauritania (MR), Niger (NE), Nigeria (NG), Senegal (SN), Sierra Leone (SL) and Togo (TG).

Similar searches were conducted using additional resources, including the UN Official Documents System, Open Grey, AGRIS FAO and Google Scholar. Searches were run on 2022-05-01.

For our analyses, we included studies that met all of the following inclusion criteria:

1. The reported findings from trapping studies where the target was a small mammal.
2. They described either the type of trap used, or the length of the trapping activity, or the location of the trapping activity.
3. They included a trapping activity from at least one West African country.
4. They recorded the genus or species of trapped individuals.
5. They were published either in peer-reviewed journals, or as preprints on a digital platform, or as a report by a credible organisation.

We excluded studies that met any of the following exclusion criteria:

1. They reported data that were duplicated from a previously included study.
2. No full text was available.
3. They were not available in English.

One author of this paper screened titles, abstracts and full texts against the inclusion and exclusion criteria. At each stage, title screening, abstract screening and full text review, a random subset (10%) was reviewed by a different author of this paper. The included_studies.xlsx in Zenodo contains the year of publication, name of the first author, title of the study, publication and unique identifier of the included studies [15].

Data were extracted from eligible studies using a standardised tool that was piloted on five randomly selected studies. The README.md contains the variable names and descriptors that were abstracted into three sheets [15]. The first sheet, "Study data", contains information on the included study, the purpose of the study, and the methodologies for rodent sampling and species identification. The second sheet, "Rodent data", contains information on the number of individuals of each species detected at a trapping location, alongside geographic coordinates of the sampling location and habitat type. The data for this section were expanded by adding non-detections if the rodent species was detected at other sampling sites within the study. Finally, the third sheet, "Pathogen data", contains information on testing the individual rodent species for known and suspected zoonotic pathogens. Unprocessed data was archived in a Zenodo repository within the 'data_raw' folder [15].

## Data validation and quality control

Species identification was assumed to be accurate in the included studies. For studies reporting genus level or multiple possible species names for a single trapped individual, data were extracted as presented in the study. Species names were mapped to GBIF taxonomy to resolve changes in taxonomic classification using the 'taxize' package (version 0.9.98) in the R statistical programming language (version 4.1.2, RRID:SCR_001905) [16, 17].

The geographic locations of trapping studies were extracted using GPS locations for the most precise location presented. Missing locations were found using the National Geospatial-Intelligence Agency GEOnet Names Server based on the placenames and maps presented in the original studies [18]. All locations were converted to decimal degrees in the EPSG:4326 coordinate reference system.

For the studies that included data, we extracted information on all microorganisms and known zoonotic pathogens tested, including the method used (e.g., molecular or serological diagnosis). Where assays were able to identify the microorganism to its species level, this information was recorded; however, for non-specific assays, higher order attribution was used (e.g., family level). For studies reporting summary results, all testing data were extracted. However, this may introduce double counting for individual rodents, for example, if a single rodent was tested using both molecular and serological assays. Where studies reported indeterminate results, these were also recorded.

We included data from preprints identified during our systematic search, in addition to studies conducted at the same locations over multiple periods. We reviewed all the occurrence data to check that the geographic coordinates, sampling period and number of identified individuals were unique, thus ensuring we did not include duplicated data. Where duplicated data were identified, we retained the record with the greatest number of detections or the most recent one. For example, multiple published studies may have included updates of a longitudinal sampling design. In these cases, only the most recent location data were retained.

## Data processing and exploration

The R code to process the raw data into the Darwin Core format for rodent occurrence, associated pathogen detection and metadata was archived as a Zenodo repository [19].

An R Shiny (RRID:SCR_001626) web application was produced to visualise the data contained in this release. The web-based application is available via a Shiny app [20]. This application allows us to explore the location of sampling sites for both rodents and their pathogens within the included studies alongside the sampling efforts reported by the studies.

## Reuse potential

This dataset of harmonised rodent species' detections, which were obtained from rodent trapping surveys conducted across West Africa, will contribute to understanding rodent biodiversity across the region. It is envisaged that this dataset will be of particular interest to researchers investigating the risk of rodent-associated zoonotic-pathogen outbreaks and emergence in this region and beyond. This data will expand the geographical coverage of the occurrence data from GBIF for most of the rodent species detected in the included rodent trapping studies with additional non-detection data of these species across the region. Where possible, dates of rodent sampling have been included. This may help researchers investigate how occurrence patterns of rodent species may vary over time, an important factor in understanding changes in the context of climate, land use and population.

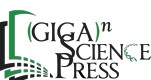

## DATA AVAILABILITY

As part of this series of Data Release articles, this dataset is available in GBIF under a CC0 1.0 Universal license [21]. Unprocessed data and the R scripts used to process it are also available from Zenodo [15, 19].

## EDITOR'S NOTE

This paper is part of a series of Data Release articles working with GBIF and supported by TDR, the Special Programme for Research and Training in Tropical Diseases hosted at the World Health Organization [22].

## ABBREVIATIONS

GBIF, Global Biodiversity Information Facility; IUCN, International Union for Conservation of Nature; UN, United Nations.

## DECLARATIONS

### Ethics approval and consent to participate

The authors declare that ethical approval was not required for this type of research.

### Competing interests

The authors declare that they have no competing interests.

### Authors' contributions

DS: conceptualisation (equal), data curation (equal), formal analysis (lead), methodology (lead), software (lead), visualisation (lead), writing original draft (lead), review and editing (equal). LAA: data curation (equal), review and editing (equal), validation (lead). KEJ: conceptualization (equal), supervision (equal), writing original draft (supporting), formal analysis (supporting), review and editing (lead). DW-J: funding acquisition (equal), supervision (equal), review and editing (equal). RK: funding acquisition (equal), supervision (equal), review and editing (equal).

### Funding

DS was supported by a PhD award from the UK Biotechnology and Biological Sciences Research Council [BB/M009513/1]. LAA was funded by a PhD award from the QMEE CDT, funded by NERC grant number [NE/P012345/1]. KEJ was supported by the Ecosystem Services for Poverty Alleviation Programme, Dynamic Drivers of Disease in Africa Consortium, NERC grant number [NE-J001570-1]. DW-J received support from the PREVAC-UP, EDCTP2 programme supported by the European Union [RIA2017S-2014]. DS and RK are members of the Pan-African Network on Emerging and Re-emerging Infections (PANDORA-ID-NET) funded by the European and Developing Countries Clinical Trials Partnership the EU Horizon 2020 Framework Programme for Research and Innovation [RIA2016E-1609].

### Acknowledgements

We would like to thank the Health data team at the Global Biodiversity Information Facility for helpful guidance in formatting our data.

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
