## [Editor Report]

Editor’s AssessmentRodents serve as hosts for various zoonotic pathogens (e.g. Lassa virus, Leptospira, etc.). However data on rodents and their pathogens suffers from taxonomic and spatial biases. To help fill that gap, the authors of this work synthesised data from 127 rodent trapping studies conducted in 14 West African countries between 1964 and 2022. The analysis of this was presented in a previous publication, but here the data was harmonised, standardised and shared with the GBIF biodiversity database. A lot of curation was required over the review process to process the raw data into the Darwin Core format, but that is now complete and available for re-use. Allowing researchers investigating the risk of rodent-associated zoonotic-pathogen outbreaks and emergence in Africa and beyond to better understand changes in rodent biodiversity.

---

## [Reviewer Report]

Comments on revised manuscriptThe revised manuscript is acceptable.  There are some issues with the GBIF dataset: 1. In the upload file, the "license" for each record is "CC-BY 4.0", but the GBIF description is "CC0". Please double-check the open license for the data. 2. The download link https://registry.nbnatlas.org/archives/dr3000/dr3000.zip is forbidden from access even when I log in. 3. There are some known issues from GBIF: 20,539 Continent derived from coordinates; 201 Country coordinate mismatch; 38 Taxon match none; 5 Taxon match fuzzy. Please fix them before publication.

---

## [Reviewer Report]

Comments on revised manuscriptThe data description is well written and the need for the data is well-motivated. Indeed, the data will make an important contribution to rodent and disease ecology. I just have a few comments. My assessment is only based on the manuscript and not on the actual datafiles   1. The title is misleading and insinuates that the authors have compiled a global database.  2. The authors claim that “Rodent trapping studies are also taxonomically and spatially biased”. The authors need to support this statement with references and be more specific about the type of biases that they see. 3. The authors motivate the data compilation with the need to understand human infections originating from rodent-borne pathogens. In this context, it would be fair to refer to the recent study by Ecke and colleagues where they show in a global study the role of synanthropic behaviour of rodent reservoirs for human risk (Ecke, F., B. A. Han, B. Hörnfeldt, H. Khalil, M. Magnusson, N. J. Singh and R. S. Ostfeld (2022). "Population fluctuations and synanthropy explain transmission risk in rodent-borne zoonoses." Nature Communications 13(1): 7532.) 4. Methods: The authors included data from preprints. How did they assure that data were not duplicated among the different data sources? 5. I suggest the authors name the countries in West Africa that were included in the study.

---

## [Reviewer Report]

Comments on revised manuscriptI have been invited to perform a data audit on the GBIF dataset associated with this manuscript, and my review will only cover that aspect of the work.  The paper outlines the digitisation of rodent occurrence data in collected in West Africa and published in scientific articles over the last few decades. It includes 127 published studies, covering an 14 (or 17? see concerns below) countries in West Africa, over the period 1964 and 2022.   Major comments (Author action required): 1 - https://registry.nbnatlas.org/archives/dr3000/dr3000.zip - Is currently not accessible (forbidden) (this may not be an issue if access is only meant to be via GBIF?)   Minor comments (Author action suggested): 1 - MS abstract states 14 countries in West Africa, GBIF dataset lists 17 different country codes? Please double check the details, and correct the MS as appropriate. (this maybe related to point below)  2 - There appears to be a mismatch between the COUNTRY name given and the location not falling within that countries (current) borders in a number (201) of instances. Please check the coordinates are correct, and adjust the country name if required.  3 - There are 38 Taxon match issues notified by GBIF automatic checks, these mostly appear to be where no species name was given, usually just “paracite”, please check and confirm that is the reason for all instances.  4 - Please ensure at least one of the GBIF contact authors is assigned the role of “Metadata provider”.  Note - The system is not allowing me to upload the full data audit report document, but it is available to GigaByte staff from here: https://drive.google.com/file/d/11K_x1JwpPbJYt64Rj5L2YEURtPJW-qsK/view?usp=sharing

---

## [Reviewer Report]

Comments on revised manuscriptThe author did a great (and swift!) job updating the data format as requested. The data format appears to be correct and nearly ready to publish via the IPT. The only adjustment necessary is to update the Kingdom designation for viruses. Viruses are a recognized kingdom in the GBIF taxonomic backbone (see https://www.gbif.org/species/8), thus this should be the designation rather than 'incertae sedis'.

---

## [Reviewer Report]

Upload additional filesDRR-202303-01/form/dSimons_vectorCompilation_mapping.xlsx.docxReviewer name and names of any other individual's who aided in reviewer Kate IngenloffDo you understand and agree to our policy of having open and named reviews, and having your review included with the published papers. (If no, please inform the editor that you cannot review this manuscript.)YesIs the language of sufficient quality?YesPlease add additional comments on language quality to clarify if needed
Are all data available and do they match the descriptions in the paper? YesAdditional CommentsAre the data and metadata consistent with relevant minimum information or reporting standards? See GigaDB checklists for examples <a href="http://gigadb.org/site/guide" target="_blank">http://gigadb.org/site/guide</a>YesAdditional CommentsIs the data acquisition clear, complete and methodologically sound?YesAdditional CommentsIs there sufficient detail in the methods and data-processing steps to allow reproduction?YesAdditional CommentsIs there sufficient data validation and statistical analyses of data quality? YesAdditional CommentsIs the validation suitable for this type of data?YesAdditional CommentsIs there sufficient information for others to reuse this dataset or integrate it with other data?YesAdditional CommentsAny Additional Overall Comments to the AuthorRecommendationMajor Revision